# Application of Transformer Models to Landslide Susceptibility Mapping

**DOI:** 10.3390/s22239104

**Published:** 2022-11-23

**Authors:** Shuai Bao, Jiping Liu, Liang Wang, Xizhi Zhao

**Affiliations:** 1School of Geomatics, Liaoning Technical University, Fuxin 123000, China; 2Chinese Academy of Surveying and Mapping, Beijing 100036, China

**Keywords:** data-driven, feature extraction, landslide susceptibility mapping, Vision Transformer

## Abstract

Landslide susceptibility mapping (LSM) is of great significance for the identification and prevention of geological hazards. LSM is based on convolutional neural networks (CNNs); CNNs use fixed convolutional kernels, focus more on local information and do not retain spatial information. This is a property of the CNN itself, resulting in low accuracy of LSM. Based on the above problems, we use Vision Transformer (ViT) and its derivative model Swin Transformer (Swin) to conduct LSM for the selected study area. Machine learning and a CNN model are used for comparison. Fourier transform amplitude, feature similarity and other indicators were used to compare and analyze the difference in the results. The results show that the Swin model has the best accuracy, F1-score and AUC. The results of LSM are combined with landslide points, faults and other data analysis; the ViT model results are the most consistent with the actual situation, showing the strongest generalization ability. In this paper, we believe that the advantages of ViT and its derived models in global feature extraction ensure that ViT is more accurate than CNN and machine learning in predicting landslide probability in the study area.

## 1. Introduction

After 2010, geological disasters have frequently occurred in many regions of China owing to global ecological changes, extreme weather conditions, and urbanization. According to the Ministry of Natural Resources of the People’s Republic of China, 18,793 geological disasters occurred between 2019 and 2022 in China. More specifically, 11,365 landslide disasters occurred, which accounted for 60.47% of all geological disasters [1]. Landslide hazards greatly threaten the productivity and livelihoods of people. The prevention and mitigation of landslide hazards have become a key concern for scholars and government departments [2,3].

Landslide susceptibility mapping (LSM) employs an evaluation model for analyzing and predicting the spatial distribution and probability of occurrence of landslide hazards [4]. It can provide a basis for spatial landslide management and policymaking with respect to disaster prevention and mitigation, and land resource planning [5,6,7,8].

Based on different theoretical foundations, LSM methods can be classified into knowledge-driven and data-driven methods [9]. Knowledge-driven LSM approaches include fuzzy logic [10], hierarchical analysis [11], and expert scoring methods [12,13]. These methods are essentially mechanical models that predict the probability of landslides occurring within a certain spatial domain based on slope stability and landslide-triggering mechanisms [14]. They are characterized by their simplicity, operability, and ability to adequately express the inherent mechanisms of landslide hazards [15]. However, compared with data-driven models, they rely more on a priori knowledge, are less objective and are better suited to the analysis of a relatively small study area. Data-driven models combine landslide conditioning factors in a weighted manner to explore the non-linear relationship between landslide hazards and landslide conditioning factors for historical landslide records of the study area [16]. Data-driven methods include classical statistical methods, such as the weight-of-evidence and informative methods and artificial intelligence methods [17], such as machine learning (ML) and deep learning (DL) algorithms [18,19,20,21,22]. Data-driven methods rely less on a priori knowledge and are more objective than knowledge-driven methods. They can handle high-dimensional large datasets, which makes them suitable for analyzing large study areas [23]. They also have a strong generalization ability. With contemporary advances in artificial intelligence theory and technology, an increasing number of scholars have successfully adopted artificial intelligence methods to build LSM models.

Among the data-driven approaches for LSM based on artificial intelligence, ML methods such as logistic regressions [24], decision trees [25], support vector machine (SVM) [26,27], random forests [28,29], and artificial neural networks have been extensively implemented [30]. Convolutional neural networks (CNN) extract image features by convolution [31]; in the CNN network structure, pooling layers reduce the amount of data while retaining useful information, and fully connected layers obtain activation values. CNNs can be used in the field of LSM as the landslide conditioning factors are stacked and formally consistent with RGB images [32,33,34,35,36]. Recurrent neural networks (RNN), another important development in DL, combine landslide conditioning factors as serialized data in LSM [37]. Nodes between hidden layers of the RNN are no longer disconnected but connected.

The development of LSM is not only affected by the innovation of classification methods but also affected by the construction of sample sets and features. The data in the form of landslide adjustment factors are mapped to higher dimensions through encoding, convolution, and classification processes, thereby improving classification accuracy [38]. Some scholars have used unsupervised learning methods [39], such as Self-Organizing Maps [40], Fuzzy C-means [41], and slope stability methods, to objectively and efficiently select non-slippery slope sample datasets [42]. This makes landslide datasets more balanced and reasonable in terms of positive and negative samples and optimizes the LSM process in terms of data content. Overlaying the hybrid, and weighted average integration of the underlying classification methods can also improve the classification accuracy of the model. CNN feature extraction can be combined with ML and RNN methods by formatting the fully connected layer data as a one-dimensional vector. This essentially adds feature extraction to ML and RNN methods to improve classification accuracy [33,38,43].

In conjunction with current research advances, ML and RNN models do not include spatial feature extraction [44,45,46]. Being geographical elements, landslides follow the “first law of geography”. Landslides are not only affected by the geography of their location but also by the geography of their surroundings. Therefore, it is crucial to extract the spatial features of landslide sites alongside those of their surroundings. CNN models use convolutional kernels to extract features from a fixed range of data, which can also be referred to as local feature extraction. Vision Transformer (ViT) models, using the encoder from a Transformer Encoder base model, have achieved good results in the field of image classification [47,48,49]. Feature extraction with ViT differs from CNN in that there is more similarity between the representations obtained at shallow and deep layers. This type of feature extraction retains more spatial information and is, therefore, referred to as global feature extraction. Swin Transformer (Swin) integrates the mobile window idea of CNN on the basis of ViT. Swin can also be used for hierarchical feature extraction similar to CNN and has a multi-scale concept at the same time.

In previous studies, two-dimensional data for a single landslide conditioning factor were formed by taking the landslide point as the center and expanding it in all directions. Multiple landslide conditioning factors are superimposed to form three-dimensional data, and disaster data are similar to RGB image data in terms of data dimensionality. ViT and its derived models also have the capability to process image data. Therefore, it is also theoretically feasible to apply ViT to the field of LSM. To date, the application of ViT and its derived models in LSM has not been documented.

This study examines the feasibility of applying ViT and its derived models in the field of LSM and analyzes the advantages of ViT in spatial feature extraction. We use a deep learning evaluation metric and the spatial distribution of known landslide points to test our proposed hypothesis. The aim was to improve the accuracy of LSM models and to provide a basis for predicting and analyzing the likelihood of landslide hazards. Meanwhile, we selected the ML (SVM), CNN (ResNet) models, which were applied together with ViT-B/16 and Swin models for LSM. The main objectives were as follows: (1) We analyze and verify the feasibility of Transformer models applied to LSM; (2) we compare and analyze the differences between different types of feature extraction methods in the LSM field.

## 2. Study Area and Data

### 2.1. Study Area

Pingwu County (31°59′ N–33°02′ N 103°50′ E–104°58′ E), a region prone to landslides, is located in the northwestern part of Mianyang City, Sichuan Province, China (Figure 1). It occupies a total area of 5950.12 km². The county is also situated in a mountainous area surrounding a basin. The mountains in its territory comprise the Minshan Mountain Range running north-south, the Motianling Mountain Range running east-west, and the Longmen Mountain Range running northeast-southwest; the elevation of 94.33% of the mountains exceeds 1000 m. The terrain is elevated in the northwest and low-lying in the southeast. The northwest area is an alpine region that gradually transitions into low-lying mountains in the southeast. The maximum elevation is 5440 m and the minimum is 600 m. Pingwu is positioned in the upper reaches of the Ful River, which is part of the Jialing River system in the Yangtze River basin. The total forestry land use area in Pingwu is 486,100 ha, accounting for 81.61% of the area under its jurisdiction, signifying a forest area of 460,000 ha and a forest coverage rate of 77.46%. Pingwu has a northern subtropical monsoon climate, with annual precipitation of 1097.2 mm and an average annual temperature of 15.0 °C.

### 2.2. Data Sources

Landslide data for the study area were obtained from the Resource and Environmental Science and Data Centre of the Institute of Geographical Sciences and Resource Environment, Chinese Academy of Sciences. The data parameters included name, number, administrative area, latitude, longitude, groundwater type, tectonic site, hazard level, and control recommendations. There was a total of 150 landslide samples, and each was mapped as a point element. The landslide data were supplemented and calibrated with satellite imagery, government reports, Google Earth images, and historical landslide studies. The landslide samples were mainly located along the Longmenshan Fault Zone from southwest to northeast. These samples had smaller widths and higher densities than other landslide samples located in the intermediate terrain and near rivers.

The slope reflects the steepness and gentleness of the surface unit, and the slope aspect determines the light intensity of the surface unit, which has an important impact on the soil moisture, vegetation type and vegetation coverage; the section curvature and the plane curvature reflect the complexity of the terrain. The stratigraphic conditions of the slope are the material basis for the occurrence of landslides. Different types of rocks composing the slope have different shear strengths and different degrees of difficulty of landslide occurrence. Normalized difference vegetation index (NDVI) can reveal the vegetation coverage in the study area, and vegetation coverage affects soil and water conservation capacity. The rainfall causes the soil-rock layer on the slope to be saturated, which increases the mass of the sliding mass and reduces the shear strength of the soil-rock layer. The river has an erosive dynamic effect on the slope body. When the water level changes, the slope body may be affected by floating, erosion, softening and weight gain. Earthquakes will induce secondary disasters such as landslides, collapses and debris flows. The main central fault zone of Longmen Mountain is along the Wenchuan-Anxian-Beichuan-Pingwu line. The landslide surface density is greater than 50%, and the maximum can reach 70%. Affect the probability of landslide occurrence.

The slope, aspect, plane curvature, and profile curvature were all generated based on digital elevation model (DEM) data. DEM data and fracture zone data were collected from the Chinese Academy of Surveying and Mapping Sciences. The spatial resolution of the DEM data was 10 m. Landslide data (https://www.resdc.cn/data.aspx?DATAID=290, accessed on 1 March, 2022) and lithology data (https://www.resdc.cn/data.aspx?DATAID=307, accessed on 1 March 2022) were obtained from the Resource and Environment Science and Data Centre of the Chinese Academy of Sciences. The rainfall data were acquired from the Chengdu Institute of Mountain Hazards and Environment, Chinese Academy of Sciences. This institution calculated the average rainfall data from 1991 to 2020 with a spatial resolution of 30 m. The landcover data were gathered from the China Earth Big Data Science Project data website (https://data.casearth.cn/sdo/detail/5fbc7904819aec1ea2dd7061, accessed on 1 March 2022) [50,51]. The temporal resolution of the data was 5 years, and the spatial resolution was 30 m. NDVI data were acquired from the China National Ecological Data Centre for Ecological Sciences (http://www.nesdc.org.cn/sdo/detail?id=60f68d757e28174f0e7d8d49, accessed on 1 March 2022). The NDVI data were based on the Google Earth Engine cloud computing platform, and all valid Landsat observations were obtained by de-clouding and de-shadowing operations [52]. Thereafter, NDVI extraction was performed for each valid Landsat observation. The landslide data and conditioning factors were unified as raster data in GeoTIFF format with a 10 m spatial resolution. We used a total of 11,879 rows and 10,665 columns of the raster for a single TIFF layer. The 11 landslide conditioning factors are shown in Figure 2.

## 3. Methods

### 3.1. Landslide Conditioning Factor Assessment

In a multiple regression model, a high correlation between two or more variables is not reasonable. In this study, the Variance Inflation Factor (*VIF*) and Tolerance Level (TOL) were chosen to quantitatively describe the correlation between the landslide conditioning factors, with TOL having an inverse relationship with *VIF* [53].
(1)$$VIF=\frac{1}{1-R_{i}^{2}}$$

*R*^2^ denotes the coefficient of determination of the *i*-th variable with all other variables; when the *VIF* of a landslide conditioning factor was >10 or TOL < 0.1, it indicated that the selected data were highly correlated, and this landslide conditioning factor was subsequently removed from the model.

Information gain can be used to analyze the weights of the characteristic factors, and it is used in LSM to determine the importance of each landslide conditioning factor. The higher the information gain rate (IGR) of the landslide conditioning factor, the more important the landslide conditioning factor is in the occurrence of landslides [54,55].

### 3.2. Support Vector Machine Model

SVM can be used to represent a convex optimization problem to discover the global minimum of an objective function using a known efficient algorithm [56]. Given a training sample set D = {(x_1_, y_1_), (x_2_, y_2_), …, (x_m_, y_m_)}, y_i_∈{−1, 1}, the most basic idea of classification learning is to find a partitioning hyperplane in the sample space based on the training set D that separates the different classes of samples. In the sample space, dividing the hyperplane can be described by the following linear equation.
(2)$$w^{T}x+b=0$$
where *w* = (*w*_1_, *w*_2_, …, *w*_d_) is the normal vector, which determines the direction of the hyperplane; *b* is the displacement term, which determines the distance between the hyperplane and the origin and the hyperplane is denoted as (*w*, *b*) (Figure 3).

### 3.3. Residual Network Model

The emergence of ResNet has solved the problem of deep network degradation to some extent [57,58]. ResNet initially aims to learn H(x), but as the number of layers of the network increases, learning H(x) becomes increasingly difficult. The learning objective thus changes to F(x) = H(x) − x, where H(x) is a layer of the network, F(x) is the residual. This learning process is called residual learning.

The ResNet50 structure used in this study (Figure 4): (i) in order to be the same size as the input data of ViT, we set the input data dimension to (224,224,11); (ii) after the first 7 × 7 convolutional layer, the output channel was 64, the step size was 2, and the pad was 3; (iii) after a 3 × 3 maximum pooling layer, the step size was 2, and the pad was 1; (iv) after conv2_x, this stage of stacking residuals was identical fast. The input size and output size were both 56 × 56 (Bottleneck down); (v) the first residual block of the conv3_x, conv4_x, and conv5_x stacks were slightly different from the other residual blocks.

### 3.4. Vision Transformer Model

The ViT model was based on the Encoder-Transformer model (Figure 5). The input image was chunked by Patch and Position Embedding in ViT. The segmented image blocks were combined into a sequence to obtain the sequence information. Then, the sequence information was passed to the Transformer Encoder for feature extraction, with the aim of adding a Classtoken to the image sequence. The Classtoken interacted with other features during the extraction process, fusing features from other image sequences. Finally, following feature extraction, the Classtoken was fully connected for classification by Multi-Head Self-Attention.

The ViT-B/16 used in this paper is the native model of ViT with a patch size of 16. There are 12 blocks in ViT-B/16. Each Block contains the multi-head attention mechanism and the MLP module. The Classtoken input is placed into 12 concatenated blocks, and the class of the input data is determined based on the Classtoken output.

### 3.5. Swin Transformer Model

Swin is a moving window ViT, which adds CNN’s multi-scale idea to its multi-head self-attention (MHSA/MSA). That is to say, the main feature of Swin is that it is structurally similar to the multi-block stack of ViT, and in terms of feature extraction, it is similar to the hierarchical feature extraction of CNN. For input data, reduce the sequence length by moving the window. The feature graph is divided into several disjoint regions, and the MHA mechanism only exists in the window. The movement of the window also makes the information exchange between adjacent windows. There is a cross-window connection between the upper layer and the lower layer, so as to achieve a global modeling capability [59].

In this paper, the Swin-T model in the Swin Transformer model is selected. The structure diagram is shown in the Figure 6. The number of blocks of Swin-T is the same as ViT-B/16, and the number of parameters is basically the same as ResNet50. Therefore, Swin-T is a reasonable choice. After the feature map goes through each stage again (except stage 1), the number of channels doubles and the width and height are halved. In the heavy classification task, Swin, like CNN, gets the classification results at the end of the model through layer norm, average pool and full connection. The Swin-T structure was used in this study (Figure 6).

### 3.6. Construction of Landslide Susceptibility Mapping Model

In the binary classification task, a comparable number of positive and negative samples is required. In this study, a buffer zone analysis was performed on 150 landslide points within a buffer radius of 2 km. One hundred and fifty non-landslide points were also randomly sampled outside the buffer zone and study area, and the non-landslide points needed to be more than 1 km apart. The dataset used in this study comprised both these landslide and non-landslide points. The spatial location data of the landslide points as well as those of the 11 landslide conditioning factors from the data centers were extracted. These data were combined into a column vector, with the data dimension applied to the SVM model. With the spatial location of the landslide as the center, location data of the 11 landslide conditioning factors were extracted with a data dimension of 224 × 224; the 11 data dimensions of 224 × 224 size were then superimposed into a three-dimensional (224 × 224 × 11) data matrix. This data dimension was applicable to the ResNet50 model, ViT-B/16 model and Swin-T model. The data extraction process is shown in Figure 7. The slice edge size was 224, considering that the size of ViT-B/16, Swin-T and ResNet50 on the ImageNet dataset was 224 × 224 × 3. The spatial resolution of the GeoTIFF format data in this paper was 10 m, which means that spatial features around the landslide point would have been extracted for about 1000 m. Therefore, the input dataset of 224 × 224 × 11 for the ResNet50, ViT-B/16 and Swin-T models in this study was spatially meaningful and feasible in a classification model.

The landslide dataset categories were used as classification labels, and landslide conditioning factors corresponding to the landslide dataset were used as classification data. We split the dataset into a training dataset, a validation dataset and a test dataset in the ratio of 60%:20%:20%. The training and validation datasets are cross-validated four-fold to avoid being based on chance accuracy. We used the scikit-learn 1.0.2 framework to implement SVM modeling and PyTorch 1.11.0 to implement ResNet50 and ViT-B/16 modeling. We input the corresponding classification models by data dimension, recorded the optimal weights, and saved the classification models. All data from the study area were entered into each classification model using the above data format, and the probability of being classified as a positive sample was recorded. All the classification results were then aggregated into predictions in the order that they were entered (Figure 8).

### 3.7. Model Evaluation Methods

In this study, the accuracy, *F*1*-score*, Receiver Operating Characteristic curve (ROC curve), and the area under the curve (AUC) were used to determine the merit of the model. The formulae for each evaluation metric were as follows.
(3)$$accuracy=\frac{TP+TN}{TP+TN+FP+FN}$$
(4)$$TPR=recall=\frac{TP}{TP+FN}$$
(5)$$FPR=\frac{FP}{FP+TN}$$
(6)$$precision=\frac{TP}{TP+FP}$$
(7)$$F1-score=2\times \frac{precision\times recall}{precision+recall}$$
where accuracy was the number of samples correctly identified by the model divided by the total number of samples; precision was the proportion of samples identified as positive by the model that was genuinely positive; *recall* was the ratio of the number of positive samples identified by the model to the total number of positive samples; *F*1*-score* was the average of the precision and recall rates of the model; ROC Curve was the curve formed by connecting the coordinate points at different thresholds using the True Positive Ratio (*TPR*) as the vertical axis and False Positive Ratio (*FPR*) as the horizontal axis; The AUC was defined as the area under the ROC curve, and the closer the ROC curve was to the upper left corner, or the larger the AUC value, the better the classification effect [60,61].

## 4. Results

### 4.1. Analysis of the Landslide Conditioning Factor

The VIF and TOL values for each landslide conditioning factor are shown in Table 1. As noted in Table 1, the VIF value of the Rain factor was the largest at 6.7795, and the VIF value of the Distance to River factor was the smallest at 1.2175. The VIF values of all the landslide conditioning factors were less than 10, indicating that there were no strong correlations between landslide conditioning factors. The 11 landslide conditioning factors selected can be used to evaluate the landslide susceptibility of this study area.

The nature of IGR states that as the weight increases, it results in an increase in the contribution of the landslide conditioning factor to the landslide. Table 2 shows the results of the weighting analysis of the 11 landslide conditioning factors. When an IGR value was greater than zero, landslide conditioning factors all contributed to the occurrence of landslides. Among them, the IGR values of Elevation and Landcover were 0.1890 and 0.1522, respectively. This indicated that the Elevation and Landcover factors exerted the greatest influence on the occurrence of landslides. The IGR values of Aspect and Plan Curvature were 0.0088 and 0.0059, respectively. This signified that the Aspect and Plan Curvature factors had the smallest contribution to the occurrence of landslides.

### 4.2. Model Validation and Comparison

Table 3 lists the accuracy values and F1-score metrics for the four classification models. In the testing dataset, the following was found for the SVM (Accuracy = 85.00%, F1*-score* = 86.15%), ResNet50 (Accuracy = 86.67%, *F*1*-score* = 86.67%), ViT-B/16 (Accuracy = 88.33%, *F*1*-score* = 88.89%) and Swin-T (Accuracy = 88.33%, *F*1*-score* = 88.52%), which shows that the Swin-T model was the most accurate. The difference between the accuracy of the four models in the training and testing datasets was not significant. This demonstrated that no significant overfitting occurred in the three classification models. In addition, the AUC values were derived from the ROC curves, and the AUC values for all four models were greater than 0.94 (Figure 9).

### 4.3. Landslide Susceptibility Mapping

The results of the model predictions were evaluated in terms of landslide susceptibility, as shown in Figure 10. The results of the four models are similar in terms of spatial distribution and probability of occurrence. The main area with a high probability of landslide occurrence was located in the southeast of the study area, showing a southwest-to-northeast distribution. The spatial distribution pattern was consistent with the location and orientation of the Longmenshan Fault Zone, which runs through this area and extended along both sides of it. Another region with a high probability of landslide occurrence was located in the central part of the study area, namely in the main urban center of Pingwu County. In fact, several rivers run through this area, and the vegetation cover is sparse. The overall spatial distribution of landslide susceptibility prediction was consistent with the spatial distribution of landslide sites.

The LSM results were analyzed by natural interval statistics and classified into five categories according to their value: very low susceptibility, low susceptibility, moderate susceptibility, high susceptibility, and very high susceptibility. The very low susceptibility zone, low susceptibility zone, and medium susceptibility zone of the SVM model accounted for 43.17%, 14.23%, and 13.32% of the study area, respectively. The high susceptibility zone and very high susceptibility zone accounted for 15.42% and 13.86% of the study area, respectively. The very low, low, and medium susceptibility zones of the ResNet50 model comprised 52.83%, 12.61%, and 9.95% of the entire study area, respectively, and the high and very high susceptibility zones accounted for 13.52% and 11.09% of the entire study area, respectively. The very low, low, and medium susceptibility zones of the ViT-B/16 model accounted for 50.96%, 14.42%, and 12.43% of the entire study area, respectively, and the high and very high susceptibility zones constituted 11.76% and 10.43% of the entire study area, respectively. The very low, low, and medium susceptibility zones of the Swin-T model accounted for 52.54%, 14.36%, and 9.92% of the entire study area, respectively, and the high and very high susceptibility zones constituted 10.00% and 13.18% of the entire study area, respectively. The comparison shows that the total proportion of very low susceptibility zones was approximately 50%, while the proportions of each of the other sub-areas were approximately 10% (Figure 11). The natural intermittency statistics of the four models showed similar trends. While ML processes data limited to raster cells at the location of landslide points, DL processes data that includes not only the landslide points but also the geospatial conditions surrounding the landslide points. Therefore, DL methods are not as sensitive as ML for areas where historical landslide points are relatively sparse or non-existent. The difference in results is that DL is more likely to predict as low susceptibility areas, and ML will predict a higher probability of landslides occurring than DL.

The results of the natural intermittent classification of LSM and the landslide points in the study area were correlated. It was found that a greater proportion of landslide points in the high and very high susceptibility zones signified an increase in the actual effectiveness of the models. The statistical results of the four models showed (Figure 12): SVM (High = 20%, Very High = 70.33%), ResNet50 (High = 26%, Very High = 63.33%), ViT-B/16 (High = 22%, Very High = 62%) and Swin-T (High = 17.33%, Very High = 71.33%). The sum of the proportion of points occurring in the high and very high susceptibility zones exceeded 80%.

We extracted portions of the study area in which no landslides were mapped but high susceptibility values were evaluated and screened them for the presence of landslides. The comparative analysis using long-term series remote sensing imagery is shown in Figure 13. The newly identified landslides agree well with the high susceptibility areas in our LSM predictions. This joint analysis confirmed that the modeling approaches adopted in this study are accurate and suitable for LSM.

## 5. Discussion

### 5.1. Model Validation and Comparison

We plotted the trends in feature maps for each deep learning model to explain the role that each sub-module within the model plays in explaining the landslide. When we reduce the variation trend of the model feature map, it is helpful to integrate and stabilize the transformed feature map [62]. Figure 14 shows the trend of feature map variance for ResNet50, ViT-B/16, and Swin-T. In the figure, white, gray and blue represent mlp/conv, multi-head self-attention and doumsample/subsample, respectively. The feature map variance of ResNet50 increases in every conv, and the last bottleneck in each conv reaches the maximum of the feature map variance. The MSA control feature map variance in the ViT model will not continue to increase, while the mlp module control feature map variance corresponding to cnn will continue to increase. The MSA and downsample phases of Swin-T reduce the feature map variance. The variance of the Swin-T characteristic map is generally at a low level. Lower feature map variance can stabilize the transformed feature maps, thus indirectly affecting the overall efficiency of the model in the later stage. Therefore, the Swin-T model performs better in terms of accuracy and *F*1 *score*.

Figure 15 shows the noise at each stage of the model in the form of the ∆Log amplitude of the high-frequency (1.0π) Fourier transform [62]. In the figure, white, gray and blue represent mlp/conv, MSA and doumsample/subsample, respectively. As can be seen from the figure, the MSA of the ViT model decreases the amplitude continuously. Moreover, the overall trend is gradually decreasing. On the contrary, the amplitude of cnn is relatively low in the early stage and becomes larger and larger in the later stage. The MSA in Swin-T also reduces the high-frequency signal, and the overall trend is smaller. According to the robustness of ViT and CNN models and the conclusion of Fourier transform results in the literature, we can draw the conclusion that low-frequency signals correspond to MSA and high-frequency signals correspond to cnn. For an image, the low-frequency signal corresponds to the shape of the image and the high-frequency signal corresponds to the texture of the image. In this paper, we analyze that ViT pays more attention to the spatial distribution of landslide units, while conv pays more attention to the texture around landslide units.

### 5.2. Analysis of Feature Extraction Results

In the north-central part of the study area, the SVM and ResNet50 models yielded similar results, with high and very high susceptibility zones distributed in bands. In contrast, the ViT-B/16 and Swin-T models predominantly showed low and very low susceptibility zones from their evaluation of the area (Figure 16).

The ViT-B/16 and ResNet50 models are fundamentally different in terms of feature extraction. The ResNet50 model gradually expands the field of view by repeatedly convolving the information around the data, layer-by-layer, through a fixed-size convolution kernel. The ViT-B/16 model uses a Self-Attentive mechanism that allows the model to have an entire field of view, even at the bottom layer. The ViT model retains more spatial location information when passing features between different feature layers. Meanwhile, the dimensionality of feature layers differs between the various Blocks of ResNet models, and the retention of the spatial location information is problematic. The ViT model yields a global representation at a shallow layer, whereas CNN models need to propagate layers further to obtain a complete representation. Therefore, the ViT model has greater feature similarity at different depths [63,64,65].

We used central kernel alignment (CKA) to calculate the similarity of feature maps inside the ResNet50, ViT-B/16 and Swin-T models (Figure 17). From the CKA results, we can see that the similarity of various bottlenecks within each conv of the ResNet is relatively large, but the similarity of various bottlenecks among different ConVs is relatively small. The blocks of the whole ViT-B/16 model are highly similar to each other. The architecture of Swin-T is similar to Vision Transformer, but the feature extraction method is similar to cnn. Therefore, the CKA result of Swin-T is similar to that of ViT-B/16, with high similarity among all blocks. Meanwhile, the similarity of Swin-T in different stages is similar to that of ResNet50, but the overall similarity is higher than that of ResNet50. This result confirms our previous analysis. Due to the structural characteristics of ViT, a better global representation is obtained in the front segment of the model. ViT passes elements between different blocks to store more spatial location information beforehand.

Prediction results for the entire study area showed that the number of high susceptibility zones predicted by the SVM and ResNet50 models was larger than that of the ViT model. This indirectly leads to more landslides being distributed in high susceptibility zones, and such results are somewhat “deceptive”. The reality is that a fracture zone is present in the study area and that there was a fracture zone, which was similar to the morphological distribution of high susceptibility zones predicted by SVM and ResNet50 modeling. However, this local area does not have records of any landslide events, indicating that the prediction results of SVM and ResNet50 modeling slightly deviated from real-world events. The comparison across model results indicated that the generalization ability of the SVM and ResNet50 models was weaker than that of the ViT-B/16 mode. Therefore, the prediction results of the ViT-B/16 model were more in line with actual events in the field.

### 5.3. Existing Problems and Future Research

Model limitations are inevitable in LSM. In fact, it is not possible to construct a model that is absolutely the best performing: it is only possible to identify a model that works best in a specific study area. The ViT-B/16 model demonstrated a high prediction accuracy and good generalization ability, and was best adapted to the actual conditions of the study area. However, the ViT-B/16 model has deeper layers and a considerable number of parameters. The ViT-B/16 model runs at roughly the same speed as the ResNet50 model but significantly slower than the SVM model. This adds limitations to the generalization ability of the ViT-B/16 model.

Non-landslide points were selected using a buffer analysis and interval sampling, which is random in nature. The landslide data used in this paper were point data and this might limit the classification accuracy. Topographic data are an important data class in the field of LSM. Previous studies have mainly used data extracted from Digital Elevation Models (DEMs) having a resolution of 30 m, whereas this study benefited from a 10 m resolution DEM. Therefore, the LSM images in this study were smoother than those obtainable from coarser resolution models. The input data size for the ResNet50 and ViT-B/16 models was 224 × 224 × 11, but it is unknown whether 224 × 224 is the optimal window size for a single layer. In addition, the ViT model strictly limits the width and height of the input data to 224, meaning that we need to extract at least 112 raster cells around the landslide point. This indirectly limits the spatial resolution of the evaluation factor. Using a landslide evaluation factor with a low spatial resolution would result in extracting too much information around the landslide point, resulting in redundant data. These issues will be further investigated in future studies [27,66,67,68].

In this study, the feature extraction method of the ViT model works best. However, the Transformers models involved in this study are in areas of relatively gentle topographic relief, and it is still unknown whether elevation will contribute to the interpretation of landslides. We will continue to compare the advantages and disadvantages of global and local feature extraction models. We will also continue combining feature extraction and ML methods, or various other feature extraction methods, with the aim of comparing and analyzing the effectiveness of single and combined feature extraction models.

## 6. Conclusions

In this study, we use SVM, ResNet50, ViT-B/16 and Swin-T models to predict the spatial distribution and probability of occurrence of landslides in Pingwu County, Mianyang City, Sichuan Province. With reference to the actual situation in the study area, 11 landslide conditioning factors were selected for use in modeling. A number of input channels of the ViT and its derived models were modified to make the LSM compatible with Transformer models. By comparison with the pre-experimental feasibility analysis and the analysis of the post-experimental results, it could be verified that Transformer models are applicable in the field of LSM. There is more similarity between the shallow and deep representations of ViT-B/16 while retaining more spatial information. The global feature extraction advantages of ViT models over CNN and ML models make them more accurate in predicting the probability of landslide occurrences in the study area. This provides a basis for subsequent investigations on combining ViT and its derived models (Swin) with other statistical and geographical methods to predict regional landslide susceptibility. It also serves as an important reference point for disaster prevention and mitigation in Pingwu County, Mianyang City, Sichuan Province, China.

## Figures and Tables

**Figure 1 sensors-22-09104-f001:**
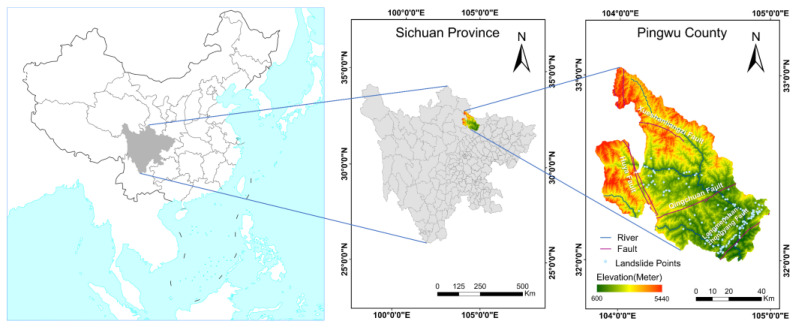
Study area and landslide distribution.

**Figure 2 sensors-22-09104-f002:**
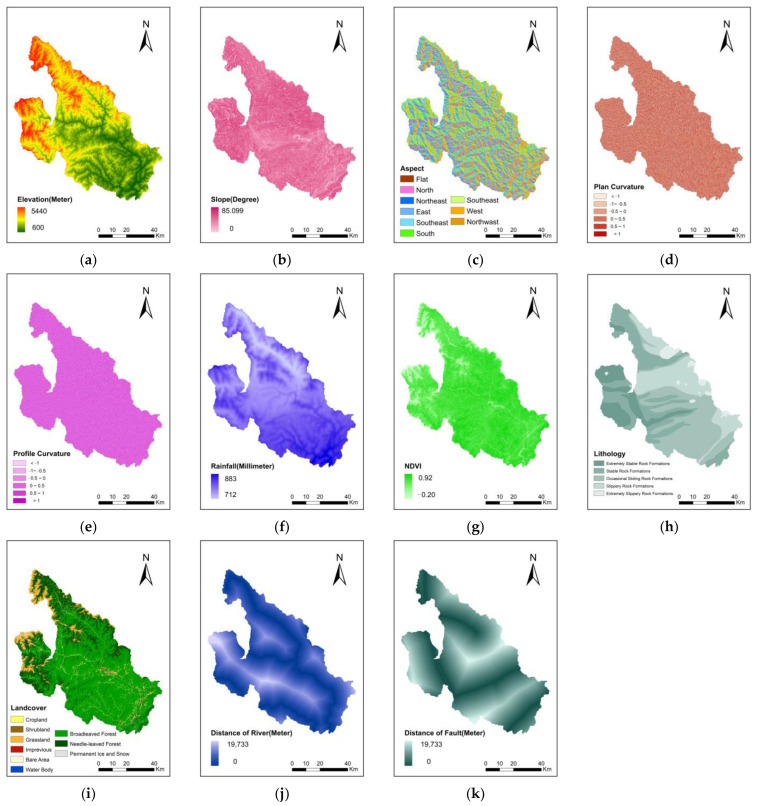
Landslide conditioning factors of LSM; (**a**) Elevation, (**b**) Slope, (**c**) Aspect, (**d**) Plan curvature, (**e**) Profile curvature, (**f**) Rainfall, (**g**) NDVI, (**h**) Lithology, (**i**) Landcover, (**j**) Distance to river and (**k**) Distance to fault.

**Figure 3 sensors-22-09104-f003:**
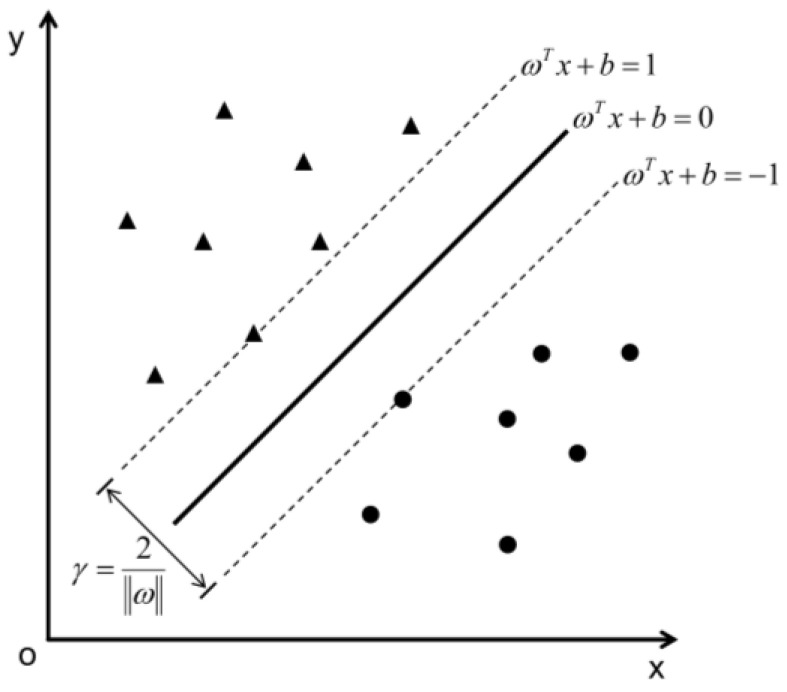
Schematic diagram of SVM.

**Figure 4 sensors-22-09104-f004:**
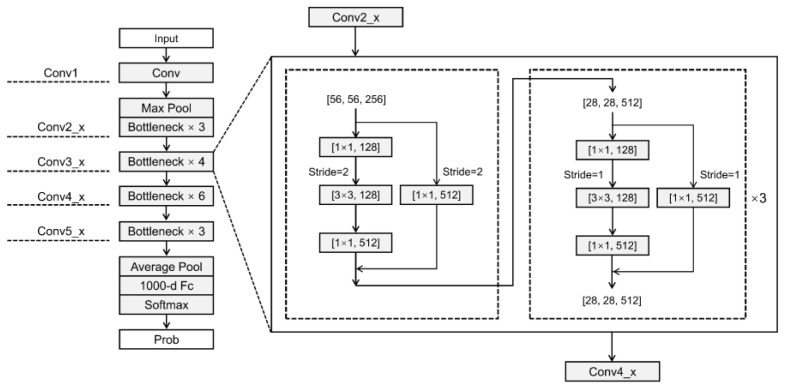
ResNet50 network structure.

**Figure 5 sensors-22-09104-f005:**
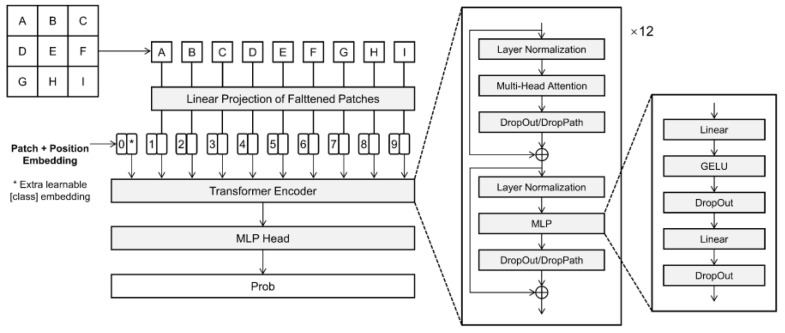
Vision transformer network structure.

**Figure 6 sensors-22-09104-f006:**
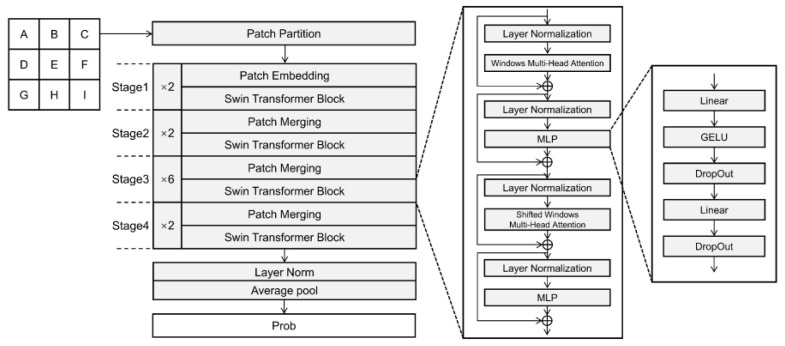
Swin transformer network structure.

**Figure 7 sensors-22-09104-f007:**
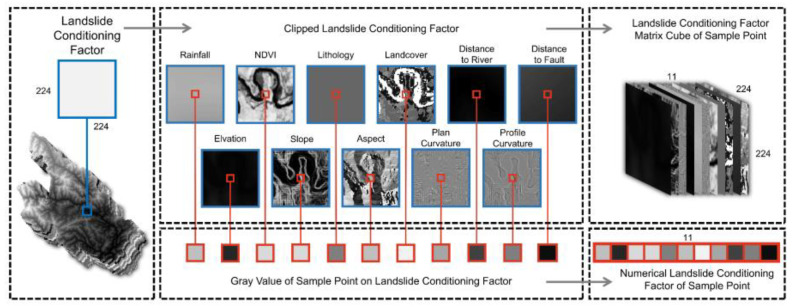
Processing of input data.

**Figure 8 sensors-22-09104-f008:**
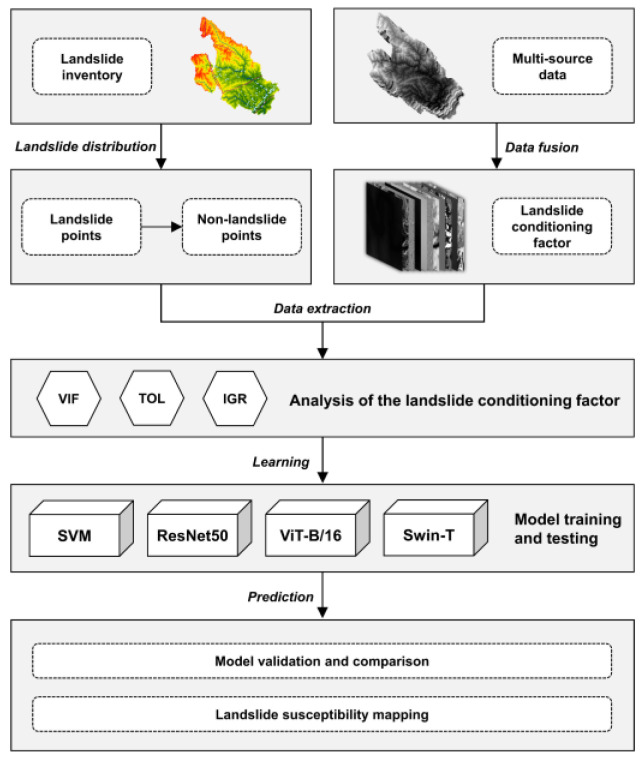
Flow chart of this study.

**Figure 9 sensors-22-09104-f009:**
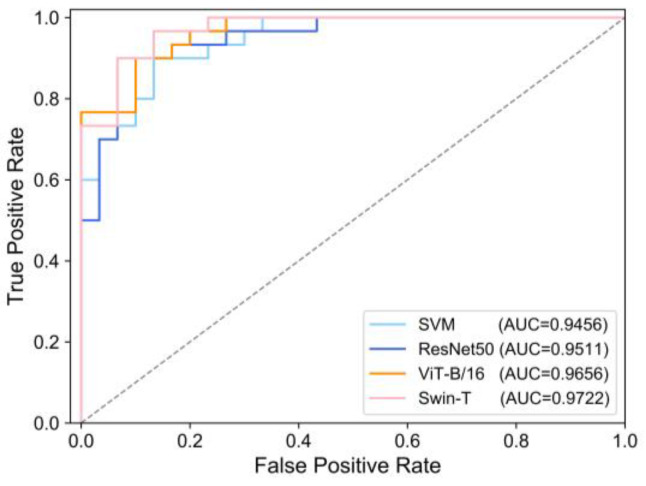
ROC curve of different LSM models based on testing dataset.

**Figure 10 sensors-22-09104-f010:**
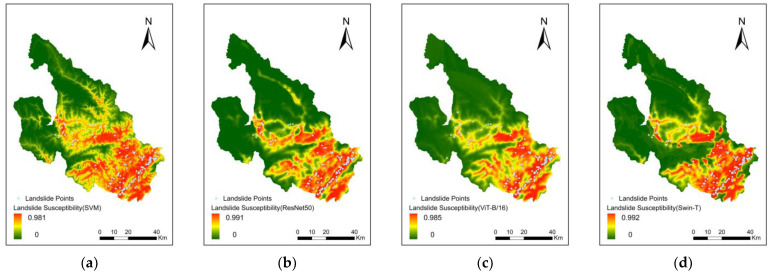
Landslide susceptibility mapping by (**a**) SVM, (**b**) ResNet50, (**c**) ViT-B/16 and (**d**) Swin-T.

**Figure 11 sensors-22-09104-f011:**
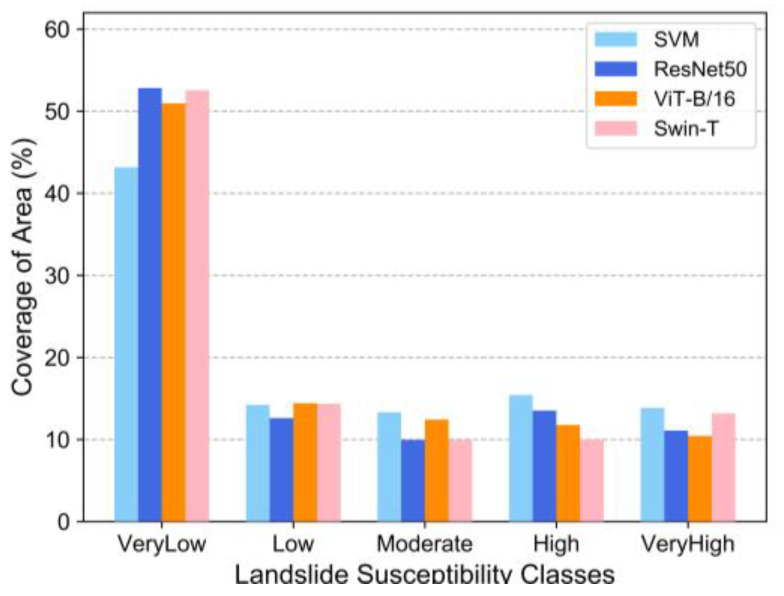
Landslide probability rating of the study area.

**Figure 12 sensors-22-09104-f012:**
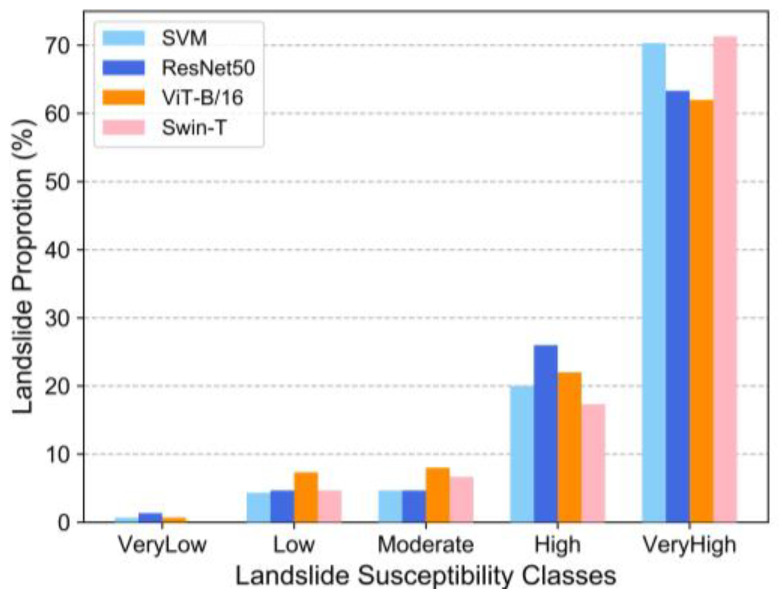
Distribution of landslide pixels in each susceptibility class.

**Figure 13 sensors-22-09104-f013:**
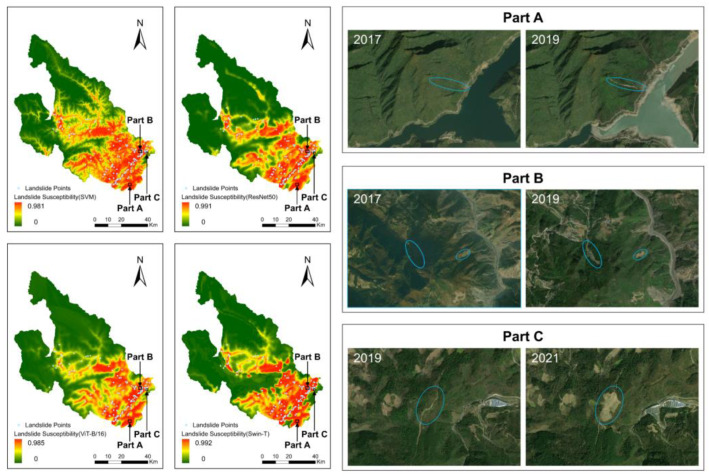
Predictive performance assessment based on remote sensing imagery.

**Figure 14 sensors-22-09104-f014:**
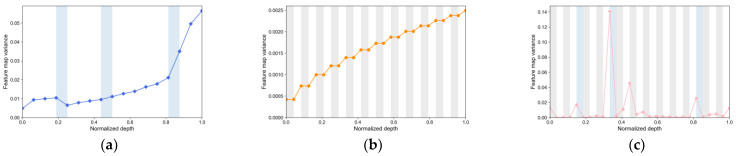
Feature map variance of (**a**) ResNet50, (**b**) ViT-B/16 and (**c**) Swin-T.

**Figure 15 sensors-22-09104-f015:**
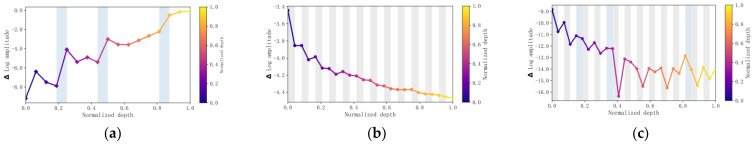
ΔLog amplitude of (**a**) ResNet50, (**b**) ViT-B/16 and (**c**) Swin-T.

**Figure 16 sensors-22-09104-f016:**
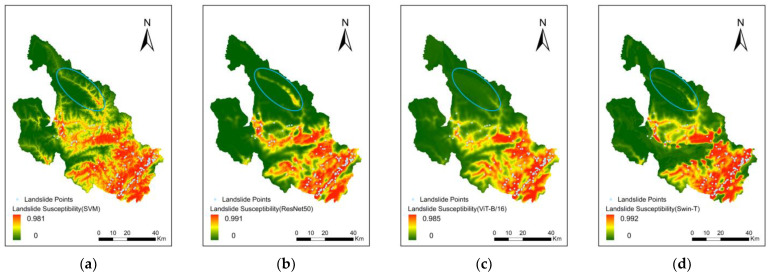
Differences in prediction results of (**a**) SVM, (**b**) ResNet50, (**c**) ViT-B/16 and (**d**) Swin-T.

**Figure 17 sensors-22-09104-f017:**
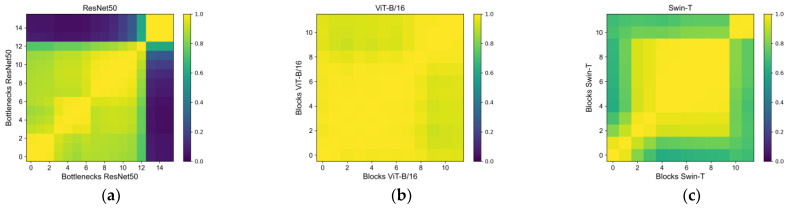
CKA values of (**a**) ResNet50, (**b**) ViT-B/16 and (**c**) Swin-T.

**Table 1 sensors-22-09104-t001:** Multicollinearity analysis results.

Factor	VIF	TOL
Elevation	1.8232	0.5485
Slope	3.4024	0.2939
Aspect	1.2796	0.7815
Plan Curvature	5.2269	0.1913
Profile Curvature	4.1913	0.2386
Rainfall	6.7795	0.1475
Lithology	2.5168	0.3973
Landcover	2.8679	0.3487
Distance of River	1.2175	0.8214
Distance of Fault	1.8096	0.5526
NDVI	4.9082	0.2037

**Table 2 sensors-22-09104-t002:** Evaluation of the role of landslide conditioning factors by IGR.

Factor	IGR
Elevation	0.1890
Slope	0.1522
Aspect	0.0953
Plan Curvature	0.0878
Profile Curvature	0.0573
Rainfall	0.0529
Lithology	0.0322
Landcover	0.0240
Distance of River	0.0115
Distance of Fault	0.0088
NDVI	0.0059

**Table 3 sensors-22-09104-t003:** Evaluation result of LSM models.

Method	Training Dataset			Testing Dataset		
	Accuracy (%)	*F*1*-Score* (%)	AUC (%)	Accuracy (%)	*F*1*-Score* (%)	AUC (%)
SVM	87.22	86.23	94.62	85.00	86.15	94.56
ResNet50	88.89	88.89	96.07	86.67	86.67	95.11
ViT-B/16	90.00	89.53	96.59	88.33	88.89	96.56
Swin-T	88.89	88.76	95.94	88.33	88.52	97.22

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
