# Peer review of "Application of Transformer Models to Landslide Susceptibility Mapping"

_sensors, 2022, doi:10.3390/s22239104_

Round 1

Reviewer 1 Report

Landslide susceptibility mapping (LSM) is of great significance. High accuracy of LSM is still needed. This draft uses Vision Transformer (ViT) to conduct LSM for the selected study area. Machine learning and convolutional neural network (CNN) model are also used for comparison. I think this work is useful for readers to better understand the methods and their features of LSM. The following comments and suggestions should be addressed. 

1 The authors compared the results of different methods and described the differences between their results. I suggest discussing the reasons for these differences. Each method has its own characteristics and advantages. One method can get good results in a specific site. It is important to tell readers about the particularities. For example, Figure 11 shows each method's accounting areas in different sensitive areas. What causes these differences?

2 This draft shows the application of transformer models to LSM. I suggest the authors to further discuss the advantages and disadvantages of this method, as well as the applicable conditions.

3 There are many abbreviations in the draft. For the convenience of reading, I suggest the authors add a table to display the full names of these abbreviations.

4 I suggest the author improve the language. I found some words are not professional.

5 I suggest the author improve the figures. The texts in some figures are too small. In addition, the name of the faults in Figure 1 should be given. A map of China is needed to show the location of the study area.

Author Response

Response to Reviewer 1 Comments

Landslide susceptibility mapping (LSM) is of great significance. High accuracy of LSM is still needed. This draft uses Vision Transformer (ViT) to conduct LSM for the selected study area. Machine learning and convolutional neural network (CNN) model are also used for comparison. I think this work is useful for readers to better understand the methods and their features of LSM. The following comments and suggestions should be addressed.

We thank the reviewer for acknowledging the value of our work, and the constructive comment for us to further improve our work. We have revised our paper to address each of the comment as documented below.

Point 1: The authors compared the results of different methods and described the differences between their results. I suggest discussing the reasons for these differences. Each method has its own characteristics and advantages. One method can get good results in a specific site. It is important to tell readers about the particularities. For example, Figure 11 shows each method's accounting areas in different sensitive areas. What causes these differences?

Response 1: In the original article, we made a partial statement about the model not being applicable in all cases “Model limitations are inevitable in LSM. In fact, it is not possible to construct a model that is absolutely the best performing: it is only possible to identify a model that works best in a specific study area.” (Page 16, Line 476-478).

In the revised version, we have added a section discussing the existence of some peculiarities in this study “the Transformers models involved in this study are in areas of relatively gentle topographic relief, and it is still unknown whether elevation will contribute to the interpretation of landslides.” (Page 17, Line 499-501).

We have also added a section to discuss the reasons for the differences between the results “While ML processes data limited to raster cells at the location of landslide points, DL processes data that includes not only the landslide points but also the geospatial conditions surrounding the landslide points. Therefore, DL methods are not as sensitive as ML for areas where historical landslide points are relatively sparse or non-existent. The difference in results is that DL is more likely to predict as low susceptibility areas, and ML will predict a higher probability of landslides occurring than DL.” (Page 12, Line 376-382).

Point 2: This draft shows the application of transformer models to LSM. I suggest the authors to further discuss the advantages and disadvantages of this method, as well as the applicable conditions.

Response 2: We have added a section to discuss some of the features of Transformer models “In addition, the ViT model strictly limits the width and height of the input data to 224, meaning that we need to extract at least 112 raster cells around the landslide point. This indirectly limits the spatial resolution of the evaluation factor. Using a landslide evaluation factor with a low spatial resolution would result in extracting too much information around the landslide point, resulting in redundant data.” (Page 17, Line 492-496).

Point 3: There are many abbreviations in the draft. For the convenience of reading, I suggest the authors add a table to display the full names of these abbreviations.

Response 3: We have kindly accepted your suggestion and added abbreviations at the end of the article. (Page 17-18, Line 531-545).

Point 4: I suggest the author improve the language. I found some words are not professional.

Response 4: We have made some necessary changes to the grammar, which is indeed an objective problem.

Point 5: I suggest the author improve the figures. The texts in some figures are too small. In addition, the name of the faults in Figure 1 should be given. A map of China is needed to show the location of the study area.

Response 5: We have zoomed in on the text in some of the figures. We have also updated Figure 1, which now includes a map of China.

Reviewer 2 Report

The study of "Application of Transformer Models to Landslide Susceptibility Mapping" provides a good novelty in susceptibility mapping, and I consider it a step forward in the field. However, some issues must be covered before acceptance. 

It appears the methodology has been designed by machine learning experts; however, the lack of cross-validation raises doubts regarding the claimed accuracy of the models.

Abstract:

Why local features should not retain spatial information for a specific case study? rewrite this part. It may not be apparent to all your readers.

study area. machine learning -->  study area. Machine learning

 Introduction:

In recent years, geological disasters... Be specific. When do authors mean?

Line 41, They are characterized by their simplicity, operability, and ability to adequately express the inherent mechanisms of landslide hazards. this needs a reference.

Also, this sentence " However, compared 42 with data-driven models, they are more dependent on a priori knowledge, less objective, 43 and only suitable for analyzing small study areas."

Line 49, Data-driven methods rely less on a priori knowledge and are more objective than knowledge-driven methods. There are some appropriate references for this statement, such as An application of Sentinel-1, Sentinel-2, and GNSS data for landslide susceptibility mapping.

Line 50, They can handle high-dimensional large datasets, which makes them suitable for analyzing large study areas. Add references like: landslide detection using deep learning and object-based image analysis.

Line 57, This part can be significantly enhanced, and my recommendation is "A convolutional neural network (CNN) extracts image features by using convolutional filters, which have been employed first for the detection of landslides [R1] and have then been applied to LSM as a result of several recent studies [R]." 

R1: https://doi.org/10.1038/s41598-021-94190-9

2. Relate works, I am not about the title of the second section.

Methods:

Line 272, The main issue of this manuscript is here: "We selected 70% of these as the training dataset and the remaining 30% as the test dataset". For any susceptibility mapping, the data set must be divided using a K-fold Cross-Validation in order to avoid based on chance accuracy. 

It is obviuse if authors use another 30%\70%, the results will be different from the SVM (Accuracy=84.44%, F1-score=83.72%), ResNet50 (Accuracy=86%, F1-score=85.71%), ViT-B/16 (Accuracy=86.67%, F1-score=86.36%) and Swin-T (Accuracy=88.89%, F1-score=89.36%).

Figure 10, the quality of the figure is not acceptable. 

Figure 13. Predictive performance assessment based on remote sensing imagery. This must be represented for all results or remove it.

Figure 14 and 15. it is not readable.

6. Conclusion

Any limitations in this work?

Author Response

Response to Reviewer 2 Comments

The study of "Application of Transformer Models to Landslide Susceptibility Mapping" provides a good novelty in susceptibility mapping, and I consider it a step forward in the field. However, some issues must be covered before acceptance.

It appears the methodology has been designed by machine learning experts; however, the lack of cross-validation raises doubts regarding the claimed accuracy of the models.

We thank the reviewer for acknowledging the value of our work, and the constructive comment for us to further improve our work. We have revised our paper to address each of the comment as documented below.

We have added cross-validation to the model training process as you suggested. In turn, the LSM and corresponding statistics for each model have been recalculated. On this basis, Figures 9, 10, 11, 12, 13, 16 and Table 3 have been updated.

Point 1: Why local features should not retain spatial information for a specific case study? rewrite this part. It may not be apparent to all your readers.

Response 1: We have rewritten this part of the Abstract “LSM based on convolutional neural networks (CNNs), as CNNs use fixed convolutional kernels, focus more on local information and do not retain spatial information. This is a property of the CNN itself, resulting in low accuracy of LSM.” (Page 1, Line 9-13).

Point 2: study area. machine learning -->study area. Machine learning

Response 2: I apologise that this was an oversight and we have corrected the spelling.

Point 3: In recent years, geological disasters... Be specific. When do authors mean?

Response 3: We have amended the first sentence of the text “After 2010, ......” (Page 1, Line 25).

Point 4: Line 41, They are characterized by their simplicity, operability, and ability to adequately express the inherent mechanisms of landslide hazards. this needs a reference.

Also, this sentence " However, compared 42 with data-driven models, they are more dependent on a priori knowledge, less objective, 43 and only suitable for analyzing small study areas."

Response 4: In response to Line 41, we have added the reference.

Application of fuzzy logic and analytical hierarchy process (AHP) to landslide susceptibility mapping at Haraz watershed, Iran. Nat. Hazards. 2012, 63, 965-996, doi:10.1007/s11069-012-0217-2.

We have re-described, the benefits of a data-driven model. The revised language is more objective and euphemistic than the original version. “However, compared with data-driven models, they rely more on a priori knowledge, are less objective and are better suited to the analysis of a relatively small study area.” (Page 1-2, Line 44-46).

Point 5: Line 49, Data-driven methods rely less on a priori knowledge and are more objective than knowledge-driven methods. There are some appropriate references for this statement, such as An application of Sentinel-1, Sentinel-2, and GNSS data for landslide susceptibility mapping.

Response 5: We have added references as you suggested.

Point 6: Line 50, They can handle high-dimensional large datasets, which makes them suitable for analyzing large study areas. Add references like: landslide detection using deep learning and object-based image analysis.

Response 6: We have added references as you suggested.

Point 7: Line 57, This part can be significantly enhanced, and my recommendation is "A convolutional neural network (CNN) extracts image features by using convolutional filters, which have been employed first for the detection of landslides [R1] and have then been applied to LSM as a result of several recent studies [R]."    R1: https://doi.org/10.1038/s41598-021-94190-9

Response 7: We have added references as you suggested.

I was surprised to find that the papers mentioned in Point 5, Point 6 and Point 7 were from the same scientific team. I then read a large number of papers by this team on landslides and felt that the papers were very well written. I have cited three more high quality papers from this team, with the following titles.

Evaluation of Different Machine Learning Methods and Deep-Learning Convolutional Neural Networks for Landslide Detection

Multi-Hazard Exposure Mapping Using Machine Learning for the State of Salzburg, Austria

Unsupervised Deep Learning for Landslide Detection from Multispectral Sentinel-2 Imagery

Point 8: 2. Relate works, I am not about the title of the second section.

Response 8: We have changed the title of section 2 to "Study area and data".

Point 9: Line 272, The main issue of this manuscript is here: "We selected 70% of these as the training dataset and the remaining 30% as the test dataset". For any susceptibility mapping, the data set must be divided using a K-fold Cross-Validation in order to avoid based on chance accuracy. 

Response 9: We sincerely accept your suggestion and have re-divided the dataset. The training dataset, validation dataset and test dataset are 60%, 20% and 20% respectively. A 4-fold cross-validation was also performed on the training and validation datasets. (Page 8-9, Line 275-277).

Based on these improved experiments, we have updated Figures 9, 10, 11, 12, 13, 16 and Table 3.

Point 10: It is obviuse if authors use another 30%\70%, the results will be different from the SVM (Accuracy=84.44%, F1-score=83.72%), ResNet50 (Accuracy=86%, F1-score=85.71%), ViT-B/16 (Accuracy=86.67%, F1-score=86.36%) and Swin-T (Accuracy=88.89%, F1-score=89.36%).

Response 10: Based on Point 9 and Response 9, we have updated the results for this section. (Page 11, Line 328-338).

Point 11: Figure 10, the quality of the figure is not acceptable.

Response 11: Based on Point 9 and Response 9, we have updated Figure 10.

Point 12: Figure 13. Predictive performance assessment based on remote sensing imagery. This must be represented for all results or remove it.

Response 12: We have modified Figure 13 by placing the full model predictions in Figure 13.

Point 13: Figure 14 and 15. it is not readable.

Response 13: We have modified and expanded the section of the text corresponding to Figures 14 and 15 with the aim of better explaining the role of this section in the text. “We plotted the trends in feature maps for each deep learning model to explain the role that each sub-module within the model plays in explaining the landslide.” (Page 14, Line 403-404) and “Figure 15 shows the noise at each stage of the model in the form of  the ∆Log amplitude of the high-frequency (1.0π) Fourier transform.” (Page 14, Line 418-420).

Point14: Any limitations in this work?

Response 14: We have revised the title of 5.3 to read 'Existing problems and future research'. At the same time, we have added two sections describing some of the problems and areas for improvement in this paper. “In addition, the ViT model strictly limits the width and height of the input data to 224, meaning that we need to extract at least 112 raster cells around the landslide point. This indirectly limits the spatial resolution of the evaluation factor. Using a landslide evaluation factor with a low spatial resolution would result in extracting too much information around the landslide point, resulting in redundant data.” (Page 17, Line 492-496) and “the Transformers models involved in this study are in areas of relatively gentle topographic relief, and it is still unknown whether elevation will contribute to the interpretation of landslides.” (Page 17, Line 499-501).

Round 2

Reviewer 1 Report

I have reviewed this draft. I think the authors have addressed all my comments and suggestions. I do not have further comments. 

Reviewer 2 Report

The manuscript is acceptable now.